# Surgical Interventions for Late Aortic Valve Regurgitation Associated with Continuous Flow-Left Ventricular Assist Device Therapy: Experience Gained and Lessons Learned

**DOI:** 10.3390/life13010094

**Published:** 2022-12-29

**Authors:** Takayuki Gyoten, Eisuke Amiya, Minoru Ono

**Affiliations:** 1Department of Cardiac Surgery, The University of Tokyo, 7-3-1 Hongo, Bunkyo-ku, Tokyo 113-8655, Japan; 2Department of Therapeutic Strategy for Heart Failure, The University of Tokyo, 7-3-1 Hongo, Bunkyo-ku, Tokyo 113-8655, Japan

**Keywords:** left ventricular assist device, LVAD explanation, weaning protocol, heart failure, mechanical circulatory support, aortic regurgitation

## Abstract

This study aimed to investigate the outcomes of surgical interventions for symptomatic moderate-to-severe aortic regurgitation (AR), including aortic valve replacement (AVR) and repair (AVP), in 184 patients who underwent continuous flow-left ventricular assist device (Cf-LVAD) implantation as a bridge-to-transplant (BTT) between November 2007 and April 2020. Ten patients (median age, 34 (25–41) years; 60% men) underwent surgical interventions (AVR, n = 6; AVP, n = 4) late after cf-LVAD implantation. The median duration after the device implantation was 34 (24–44) months. Three patients required additional tricuspid valve repair. Aortic valve suturing resulted in severe recurrent AR 6 months postoperatively, due to leaflet cutting in one patient. Seven patients with AVR survived without regurgitation during the study period, except for one non-survivor complicated by liver failure due to postoperative right heart failure. Therefore, six patients after AVP (n = 4) and AVR (n = 2) underwent successful heart transplantation 7 (4–13) months after aortic intervention. Kaplan–Meier analysis showed no significant difference in overall survival through 5 years after cf-LVAD implantation, regardless of the surgical AV intervention chosen (log-rank test, *p* = 0.86). In conclusion, surgical interventions (AVR or AVP) for patients with an ongoing cf-LVAD are safe, effective, and viable options.

## 1. Introduction

Continuous flow-left ventricular assist device (Cf-LVAD) implantation is a life-saving procedure for drug-refractory end-stage heart failure (HF) with severe left ventricular dysfunction [1,2]. In recent years, due to limited donor availability, the waiting time for heart transplantation (HTx) is becoming longer. Therefore, LVAD-associated complications have been gaining attention. Aortic regurgitation (AR) is a common cause of morbidity and mortality in patients with Cf-LVAD, because the reduction of effective LVAD forward flow causes low cardiac output syndrome [3]. A previous study reported that 25–30% of patients develop AR within one year after the device’s implantation [4]. For AR in patients with implanted LVADs, conservative treatment options, including LVAD speed adjustment, inotropic therapy, blood pressure management, or forced diuresis, might have a limited impact on circulation [5]. In patients with AR inducing uncontrolled HF, surgical interventions, such as surgical aortic valve replacement (SAVR) or repair (AVP), may be secondarily selected as a bridge-to-heart transplantation (BTT), based on the local transplant allocation systems [6]. In Japan, the recent waiting time for HTx is longer than 5 years [7]. Therefore, surgical interventions for AR remain the only feasible option for overcoming this increasingly prolonged waiting period [8]. This also applies to patients undergoing destination therapy. However, the effect of surgery on AR is poorly investigated, and there are no clear recommendations regarding surgical interventions for AR in patients with cf-LVAD [5]. In addition, the durability of surgical AVR and AVP remains unknown, and surgical procedures that may provide long-term prevention of recurrent AR and the timing for such interventions need to be investigated. Therefore, our study aimed to investigate the clinical outcomes of surgical interventions, including AVR and AVP, for symptomatic AR in patients with cf-LVAD implantation from a single center. We concluded that AVR and AVP are safe and feasible interventions for patients who have an ongoing LVAD, as well as a BTT for those with HF secondary to AR.

## 2. Materials and Methods

### 2.1. Study Design and Follow-Up

This single-center study was approved by the institutional Ethics Committee of the University of Tokyo Hospital (3031-[4]). Hospital records were screened to retrospectively identify patients with moderate-to-severe AR and cf-LVAD implantation who were treated in our hospital between November 2007 and April 2020 with surgical AVR or AVP. Several devices were used during the study period, including the DuraHeart (Terumo Heart, Ann Arbor, MI, USA), EVAHEART (Sun Medical technology Research Corp, Nagano, Japan), Jarvik 2000 (Jarvik Heart, New York, NY, USA), HVAD (Medtronic, Minneapolis, MN, USA), HeartMate II (Abbott Medical, Abbott Park, IL, USA), and HeartMate 3 (Abbott Medical, Abbott Park, IL, USA). Patients who were younger than 18 years, treated with destination therapy, or had uni-ventricular anatomy were excluded (Figure 1). Clinical course of all patients was retrospectively reviewed based on patient charts including preoperative and postoperative periods in the intensive care unit, and at the time of discharge. Follow-up data on clinical status and transthoracic echocardiography (TEE) were obtained from our databases and complete in 100% of the patients. Clinical follow-up concluded on 31 May 2021, when the last enrolled patient completed 1 year of follow-up. The study end-point was all-cause death. HF was categorized into new-onset or worsening signs and symptoms that required urgent therapy resulting in hospitalization.

### 2.2. Treatment Strategies for Aortic Regurgitation (Figure 2)

Patients were regarded as potential candidates for AVP if they met the following criteria: (1) AR originated from the center of the AV (central jet), (2) the AV comprised three leaflets, each with a wrinkly surface on one side and smooth on the other (normal anatomy), and (3) if there was difficulty in gaining the optimal surgical view of the aortic annulus for AVR (only observation of the aortic cusps was possible). The local heart team, consisting of a cardiologist, cardiac surgeon, perfusionist, and cardio-anesthetist, discussed individual therapeutic approaches based on age, surgical risk, cardiac and extra-cardiac comorbidities, and aortic valve morphology as assessed by doppler transthoracic echocardiography (TTE) and TEE. The severity of AR assessed by echocardiography was classified as none (0), trivial/mild (1), moderate (2), moderate-severe (3), or severe (4).

**Figure 2 life-13-00094-f002:**
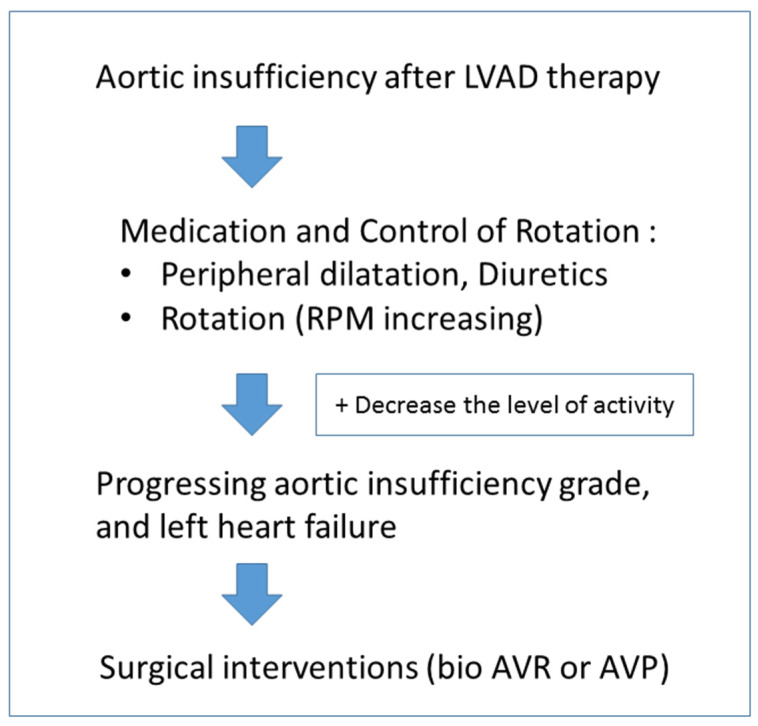
Approach to aortic insufficiency in patients with LVAD. LVAD, left ventricular assist device; RPM, rotations per minute; AVR, aortic valve replacement; AVP, aortic valve repair.

### 2.3. Surgical Procedures

All procedures were performed by experienced, board-certified cardiovascular surgeons. After performing a median re-sternotomy, cardiopulmonary bypass (CPB) was established through direct cannulation to the ascending aorta and the right atrium, or superior and inferior vena cavae in cases requiring tricuspid valve (TV) surgery. Concomitant procedures, including tricuspid annuloplasty (TAP) or TV replacement, were performed if needed. The AV was accessed via a small oblique aortotomy incision. Either standard AVR with biologic prosthesis or AVP with a central coaptation stitch (Park’s stitch suture) was performed on the arrested heart under normal temperature (36 °C) [9]. Intraoperative TEE was performed to verify the absence of AR after AVR and residual AR of grade 0–1 in AVP. All patients were postoperatively treated with warfarin for anticoagulation aiming to maintain the international normalized ratio at 2.3–2.7 as a normal routine.

### 2.4. Statistical Analysis

Continuous variables are expressed as medians (interquartile ranges [IQRs] of the 25th–75th percentiles) and categorical variables as frequencies and percentages. Univariable comparisons were performed using Student’s unpaired *t*-test for continuously normally distributed data. Mann–Whitney U test was used for comparisons of non-parametric data and Fisher’s exact test for categorical variables. Data regarding survival and freedom from all-cause death were derived using the Kaplan–Meier method; comparisons were made with a log-rank test. Statistical significance was set at a two-sided *p*-value of <0.05. All statistical analyses were performed using the R software (The R Project for Statistical Computing; The R Foundation, Vienna, Austria).

## 3. Results

### 3.1. Preoperative Characteristics

Of the 184 adult patients implanted with Cf-LVADs at our center during the study period, 156 were ultimately evaluated after applying the exclusion criteria. Due to the development of symptomatic AR (grade 2 [IQR: grades 2–4]) that progressed to uncontrolled HF, 10 patients (six men; median age: 34 [IQR: 25–41] years; except one patient who required complete AV closure) underwent AV surgery at a median of 34 (IQR: 24–44) months after the Cf-LVAD implantation (Figure 1). Five patients with combined AR and uncontrolled HF opted for conservative therapy with careful informed consent. The baseline characteristics of the 10 patients are summarized in Table 1. The previously implanted Cf-LVADs included the DuraHeart (n = 1), EVAHEART (n = 1), Jarvik 2000 (n = 3), and HeartMate II (n = 5). No patient underwent Impella (ABIOMED, Danvers, MA, USA) implantation prior to Cf-LVAD therapy. Three patients underwent concomitant surgeries of the other valves, such as TAP (n = 3) and mitral annuloplasty (n = 1), during LVAD implantation. No patient showed AR of grade two or higher at the time of the initial Cf-LVAD surgery. All patients received Cf-LVAD through median sternotomy.

### 3.2. Echocardiographic and Hemodynamic Information

The TTE and right heart catheter values are summarized in Table 1. Most patients (n = 8) had no opening of the AV with a mild-severe grade AR (grade 3 [IQR: 2–4]). The value of tricuspid annular plane systolic excursion (TAPSE) was low in patients who underwent AVP and AVR. Compared to the values observed in patients who underwent AVR, the median diameters of the ascending aorta (AsAo), sinotubular junction, sinus of Valsalva, and aortic annulus are, not significantly, decreased in patients who underwent AVP. The median right atrial pressure was 12 (IQR: 9–18) mmHg; however, four patients had values higher than 16 mmHg. The median cardiac index (CI) was 1.8 (IQR: 1.6–2.3) L/min/m^2^, and all patients had CI values lower than 2.2 L/min/m^2^, except for two with CIs of 2.7 and 2.5 L/min/m^2^, respectively.

### 3.3. Intra- and Postoperative Outcomes

Table 2 describes the intraoperative data and postoperative outcomes during the follow-up period. The durations of CPB and aortic cross-clamping in the AVP group had medians of 172 (IQR: 160–179) min and 61 (IQR: 58–70) min, respectively. The AsAo-clamp time was longer in the AVR group than in the AVP group (Table 2, *p* = 0.067). For AVR, Carpentier-Edwards Magna 21 mm (n = 2; Edwards LifeSciences LLC, Irvine, CA, USA), Crown PRT 19–21 mm (n = 3; LivaNova PLC, London, UK), and Inspiris 23 mm (n = 1; Edwards LifeSciences LLC, Irvine, CA, USA) were used. At the time of AVR, one patient required concomitant TVR, while two patients required TVR with biological prostheses in the AVP group.

During the study period, eight patients were discharged home without any further complication; one patient died due to liver failure secondary to right HF 42 days after AVR. One patient developed severe recurrent AR 6 months after AVP and required intensive medication therapy for aggravated HF secondary to severe grade AR for an additional 6 months until HTx. Overall, all four patients in the AVP group reached HTx within 6 (IQR: 4–8) months (Table 2). Two of the AVR group underwent HTx, while the other three remained on LVAD support after 9, 13, and 15 months on the waiting list, respectively.

### 3.4. Impact of Aortic Valve Interventions on Long-Term Survival

A Kaplan–Meier analysis of the enrolled patients (n = 156) showed that the freedom rate from all-cause mortality after Cf-LVAD implantation was not significantly different for surgical AV interventions (AVP, AVR, and complete AV closure) for AR-related HF at 5 years of follow-up compared with non-surgical AV interventions, including conservative therapies for both AR and non-AR (log-rank test, *p* = 0.86) (Figure 3).

### 3.5. Intraoperative Pathological Findings Associated with Aortic Regurgitation

AR was caused by various aortic valve pathologies such as prolapse (n = 2), degeneration (n = 5), or annular dilatation (n = 1) (Table 2 and Appendix A). In two patients, anatomical findings could not be observed due to an extremely limited surgical field; AVP was performed for these patients. In all three patients with a prolapsed AV, only a portion of the right coronary cusp was prolapsed. AV degeneration in the five patients showed endothelialization and myxomatous changes (Table 2). Degeneration was observed in all three coronary cusps in three patients and only in the right coronary cusp in two (Figure 4). The fusion of two cusps (i.e., fusion between the non- and left coronary cusps) was observed in three patients who underwent AVR.

## 4. Discussion

As the number of annually implanted Cf-LVADs is increasing worldwide, the overall number of patients with VAD-related complications is also increasing [10,11]. Late-onset AR affects approximately 25% of patients with LVAD within one year after the device’s implantation [4]. Rapid AR progression was reported in a few cases [3]. AR-related blood recirculation leads to both forward and backward HFs, including secondary right HF with tricuspid regurgitation in some cases [8]. Furthermore, Cf-LVAD reduces pulsatility and induces the infrequent opening or even permanent closure of the AV if the pump speed is not properly adjusted. Currently, official recommendations regarding the treatment of late-onset AR remain unestablished [12].

In Japan, the waiting time for HTx has increased to more than 5 years due to a shortage of donors [7]. Our allocation system does not include a high-urgency system, differing from the European transplant system [7,13]. Thus, AR associated with cf-LVAD is an issue that requires our immediate attention. In our cohort, surgical AV interventions for AR were performed at a median of 34 (24–44) months after Cf-LVAD therapy. Interestingly, a preoperative TTE showed a moderate-severe grade tricuspid regurgitation with a low value of TAPSE (≤14 mm) in four patients (40%). Overall, concomitant TV repair or replacement was performed for three patients, of whom one died of liver failure secondary to progressive right HF. The right ventricular (RV) function was mostly preserved in the remaining six patients during the perioperative phase.

This may be due to three potential causes: (1) once the AR-associated backward failure is corrected, LVAD increased forward flow, thereby increasing the RV preload, which might have resulted in a worsening of RV failure in case of a preexisting RV dysfunction; (2) all patients already suffered from right HF secondary to severe AR prior to surgery; and (3) cardioplegic arrest for AVR might not be protective for the right ventricle.

In such a scenario, minimally invasive strategies for AR with HF could be a treatment option; a superiority of transcatheter AVR (TAVR) over SAVR for these cases is a topic of ongoing discussion [6,14,15]. However, especially in the absence of valve calcification and with LVAD support, the risk of valve dislocation might increase [16]. Gyoten and Rojas et al. reported performing an emergent SAVR after an unsuccessful TAVR due to a dislocation of the transcatheter valve (CoreValve, Medtronic, Minneapolis, MN, USA) [8]. Actually, no confirmed procedures for a complete valve repair are recommended so far.

Despite the risks, surgical AVP and AVR with a biological prosthesis might be the best treatment options for AR. Of our cohort, 10/156 (6%) underwent AVP or AVR. However, no guidelines to select AVP or AVR exist, and the surgical strategy depends on the experience of each surgeon. From our experience, AVP is a simple surgical option for AR management, and is increasingly used for cf-LVAD, because the valve has the potential for opening during systole to clear the aortic root of stasis [17]. However, the durability of AVP remains unknown. Therefore, the risk of recurrent AR due to leaflet cutting (as in our case) should be monitored carefully during a longer follow-up period. On the other hand, no or little residual AR was observed in the AVR cohort (Table 2 and Appendix A). In cases where the optimal surgical field cannot be achieved due to hard wall thickness of the AsAo secondary to a redo cardiac surgery, bioprosthetic valve implantation may be difficult. Additionally, there is a possibility to form the membrane of fibrotic tissue on the LV side, even if AVR could be performed successfully (Figure 5). This subaortic valve-membrane may be associated with thrombus formation due to blood stasis.

### Limitations

Our study had some limitations. This was a retrospective, single-center study with a limited number of patients. Additionally, the therapy selection (AVP or AVR) depended on the surgical view, AV pathology, or concomitant procedures. Therefore, the effect of each strategy could not be compared. Although this study could not provide generalizable findings, we were still able to report good outcomes for this cohort.

## 5. Conclusions

Surgical intervention (AVP or AVR) is both safe and feasible for patients with HF secondary to late-onset AR and is a viable option for selected patients. It may be prudent to consider performing AVP or AVR before the progression to right HF.

## Figures and Tables

**Figure 1 life-13-00094-f001:**
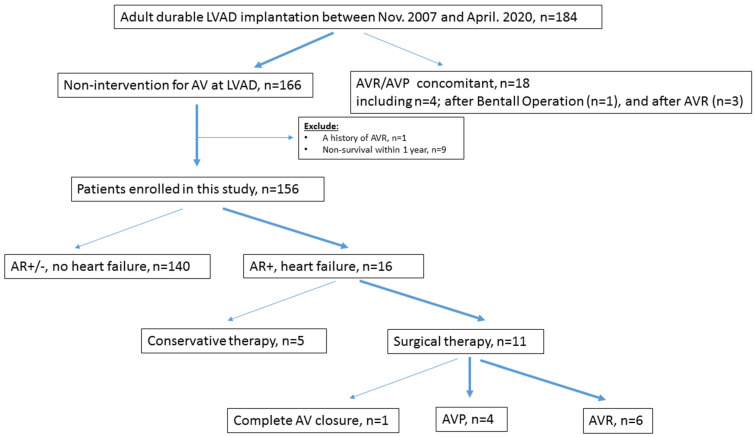
Flowchart for selection of patients. LVAD, left ventricular assist device; AR, aortic regurgitation; AV, aortic valve; AVR, aortic valve replacement; AVP, aortic valve repair.

**Figure 3 life-13-00094-f003:**
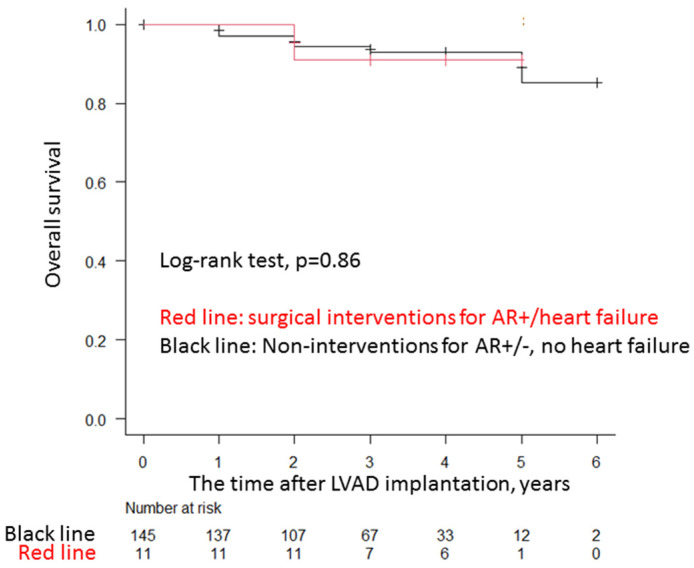
Clinical outcomes of AV interventions. Kaplan–Meier survival curves for all-cause mortality after aortic valve intervention vs. non intervention in patients with Cf-LVAD are shown (*p* = 0.86). cf-LVAD, continuous flow-left ventricular assist device; AR, aortic valve regurgitation; AV, aortic valve.

**Figure 4 life-13-00094-f004:**
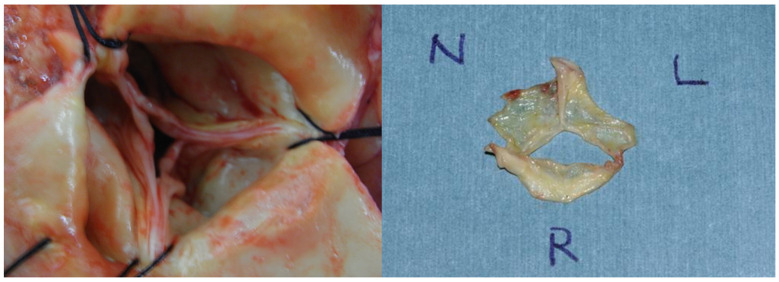
Pathology of aortic valve regurgitation in two unpublished cases. (**Left**): prolapse of one leaflet (non-coronary cusp, in this case). (**Right**): degenerative alteration of the right coronary cusp.

**Figure 5 life-13-00094-f005:**
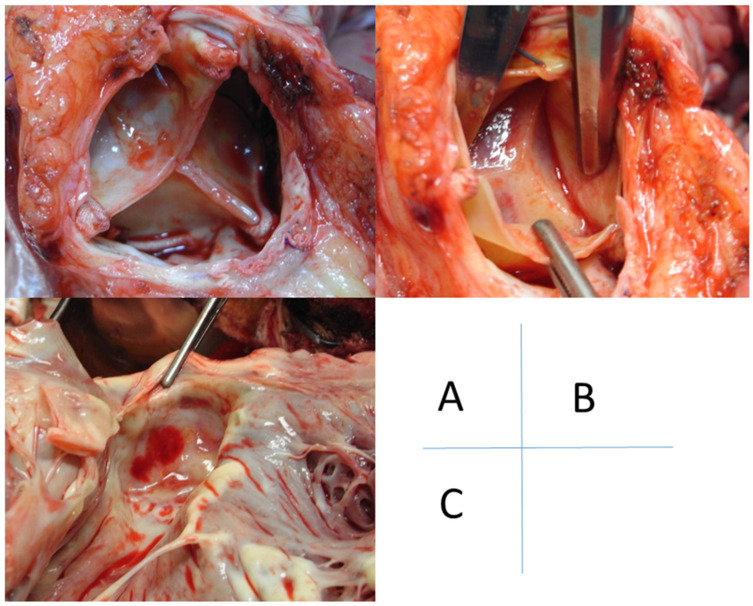
Bioprosthesis-associated alteration. A 36-year-old man underwent heart transplantation 2 years after aortic valve replacement with bovine prosthesis and durable LVAD therapy. (**A**) The bioprosthesis has no pathological alteration. (**B**) On viewing from the side of the ascending aorta, a membrane consisting of fibrous tissue is formed under the bioprosthesis. (**C**) On viewing from the side of the left ventricle, a membrane impedes the left ventricular outflow. LVAD, left ventricular assist device.

**Table 1 life-13-00094-t001:** Baseline characteristics, echocardiography, and hemodynamic data at the time of surgical intervention for aortic valves.

	Total n = 10	AVP n = 4	AVR n = 6	*p*-Value
Age at LVAD implantation, years	32 (22–38)	23 (21–26)	37 (32–40)	0.13
Age at aortic valve intervention, years	34 (25–41)	26 (25–28)	40 (34–44)	0.11
Male	6	2	4	1
Body mass index, kg/m^2^	19 (18–20)	18 (17–19)	20 (19–20)	0.35
Body surface area, m^2^	1.6 (1.5–1.8)	1.5 (1.4–1.6)	1.6 (1.5–1.8)	0.35
Pathology				0.77
Dilated cardiomyopathy	5	1	4	
Dilated phase of hypertrophic cardiomyopathy	3	2	1	
Arrhythmogenic right ventricular cardiomyopathy	1	1	0	
Transposition of the great arteries	1	0	1	
INTERMACS level at cf-LVAD implantation	3 (3–3)	3 (2–3)	3 (3–3)	0.087
A history of mechanical circuratory support				
Impella	0	0	0	1
IABP	1	1	0	0.33
PCPS	0	0	0	1
Nipro para-corporeal pulsatile LVAD	3	0	3	0.2
Implanted LVAD device types				1
DuraHeart	1	1	0	
EVAHEART	1	0	1	
HeartMate II	5	2	3	
HeartMate 3	0	0	0	
HVAD	0	0	0	
Jarvik 2000	3	1	2	
Concomitant TV surgery at LVAD implantation	3	1	2	1
VAD duration to aortic valve intervention, month	34 (24–44)	37 (27–44)	34 (25–42)	1
Laboratory				
Brain natriuretic hormone, pg/mL	579 (369–1039)	554 (399–799)	579 (397–1230)	0.76
Echocardiography				
Aortic regurgitation, grade	3 (2–4)	4 (3–4)	2 (2–3)	0.071
Aortic valve-opening	2	0	2	0.47
Mitral regurgitation, grade	2 (1–2)	1 (1–2)	2 (2–2)	0.3
Tricuspid regurgitation, grade	2 (2–4)	4 (3–4)	2 (2–2)	0.1
Tricuspid annular plane systolic excursion, mm	8 (7–10)	7 (7–8)	10 (8–12)	0.12
Annulus, mm	21 (18–22)	20 (18–21)	22 (19–24)	0.33
Valsalva, mm	27 (25–30)	26 (25–27)	30 (25–31)	0.45
Sinotubular junction, mm	23 (19–25)	21 (19–22)	24 (21–26)	0.28
Ascending aorta, mm	27 (25–29)	26 (25–28)	27 (26–29)	0.45
Right heart catheter after increased RPM				
Mean right atrial pressure, mmHg	12 (9–18)	11 (10–15)	14 (10–19)	0.7
Right ventricular end-diastolic pressure, mmHg	13 (9–16)	12 (10–13)	15 (9–18)	0.57
Mean pulmonary artery pressure, mmHg	18 (16–23)	16 (14–18)	21 (18–31)	0.19
Mean pulmonary capillary wedge pressure, mmHg	12 (12–18)	8 (8–10)	17 (13–23)	0.049
Cardiac output, L/min	3.2 (2.7–3.4)	3.2 (3.2–3.4)	2.9 (2.5–3.4)	0.55
Cardiac index, L/min/m^2^	1.8 (1.6–2.3)	2.3 (2.0–2.4)	1.6 (1.6–1.9)	0.26

n is expressed as median (interquartile range), if not otherwise specified. Cf-LVAD, continuous flow-left ventricular assist device; AVP, aortic valve repair; AVR, aortic valve replacement; INTERMACS, interagency registry for mechanically assisted circulatory support; IABP, intra-aortic balloon pumping; ECMO, extracorporeal membrane oxygenation; TV, tricuspid valve; RPM, rotations per minute.

**Table 2 life-13-00094-t002:** Intraoperative findings and postoperative outcomes.

	Total n = 10	AVP n = 4	AVR n = 6	*p*-Value
Cardiopulmonary bypass time, min	172 (152–191)	172 (160–179)	173 (152–205)	0.91
Ascending aorta clamp time, min	81 (66–94)	61 (58–70)	88 (80–117)	0.067
Concomitant procedures				
TAP	1	0	1	
TVR	2	2	0	
Main reason of AR				
Prolapse	2	1	1	
Degenerative change	5	1	4	
Dilatation of annulus	1	0	1	
Unknown	2	2	0	
Outcomes				
Recurrent severe AR	1	1	0	
AR grade at 1 month after surgery	0.5 (0–1)	1 (1–1)	0 (0–0)	0.018
AV-opening during follow up period	No	No	No	
Death	1	0	1, liver failure	
Explant	0	0	0	
On-going LVAD support	3	0	3	
Heart transplantation	6	4	2	
Waiting time for HTx after AV surgery, month	7 (4–13)	6 (4–8)	4 and 43	
On the waiting list, month	-	-	9, 13, and 15	

AVR, aortic valve regurgitation; AVP, aortic valve repair; AV, aortic valve; AR, aortic regurgitation; TAP, tricuspid annuloplasty; TVR, tricuspid valve replacement; LVAD, left ventricular assist device; HTx, heart transplantation.

## Data Availability

Not applicable.

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
