# Peer review of "Surgical Interventions for Late Aortic Valve Regurgitation Associated with Continuous Flow-Left Ventricular Assist Device Therapy: Experience Gained and Lessons Learned"

_life, 2022, doi:10.3390/life13010094_

Round 1

Reviewer 1 Report

A study that can lay the basis for a guideline in the follow-up of patients using LVAD and it can be published without any revision.

Author Response

Dear  Reviewer 

Thank you for your review.

Sincerely

Takayuki Gyoten

Reviewer 2 Report

This is an observation study for AR in LVAD implantation patients. It is well summarized, but I have question about treatment option.

Is the surgery only option for this situation?

TAVI is good option for this kind of situation.

And for valve replacement, sutureless or rapid deployment valve is more valuable for reducing risk.

For aortic valve repair, it is very difficult to treat aortic valve via leaflet repair procedure. In addition, no confirmed procedures for complete valve repair are recommended so far.

Authors should add additional results and comments for these issues.

Author Response

Thank you very much for your meaningful suggestions, we could revise our manuscript in total up to the very valuable and helpful suggestions of the reviewer. We corrected the next sentences as suggested.  

Q1, Is the surgery only option for this situation?

A1, Yes, only surgery is acceptable in Japan, but treatment option was choiced according to our strategy.

Q2, TAVI is good option for this kind of situation.

A2, TAVI is not common option in Moment, although some authors reported the superiority of TAVI. In Japan, TAVI is not one of surgical options. Gyoten, et al (Artificial Organ, 2021;45:736-741) reported that TAVI has a risk of emergent surgery secondary to a device migration. The option trends to cause high mortality or right heart failure caused by cardiogenic shock due to a new acute AR.

Q3, And for valve replacement, sutureless or rapid deployment valve is more valuable for reducing risk.

A3, These new valves has a high risk of AV-block than standard AVR. Simple AVR is easier and fast. Therefore we did not use sutureless or rapid deployment valve.

Q4, For aortic valve repair, it is very difficult to treat aortic valve via leaflet repair procedure. In addition, no confirmed procedures for complete valve repair are recommended so far.

A4, Yes , AV-plasty with Park Stitch technique is very easy, except leaflet fragility to treat.

C4, However, especially in the absence of valve calcification and for LVAD support, the risk of valve dislocation might increase [16].Gyoten and Rojas et al. reported performing an emergent SAVR after an unsuccessful TAVR due to dislocation of the transcatheter valve (CoreValve, Medtronic, USA) [8]. Actually, no confirmed procedures for complete valve repair are recommended so far.

Reviewer 3 Report

The authors report their experience with aortic valve surgery for AR in patients under LVAD.

The topic of the paper is novel and interesting.

It is not clear in the abstract, in the results and discussion section how many patients developed again severe AR after AVP. Please clarify.

It would be interesting to have an additional table with the type of intra operative finding, the type of intervention and the outcome (also regarding AR) for each of the ten patients.

Author Response

Thank you very much for your meaningful suggestions, we could revise our manuscript in total up to the very valuable and helpful suggestions of the reviewer. We corrected the next sentences as suggested.  

The authors report their experience with aortic valve surgery for AR in patients under LVAD.

The topic of the paper is novel and interesting.

Q1, It is not clear in the abstract, in the results and discussion section how many patients developed again severe AR after AVP. Please clarify.

A1, in one patient

C1, in one patient

Q2, It would be interesting to have an additional table with the type of intra operative finding, the type of intervention and the outcome (also regarding AR) for each of the ten patients.

A2, Yes, we added this information.

C2, Please see Table S1